# Activated Carbon Preparation from Sugarcane Leaf via a Low Temperature Hydrothermal Process for Aquaponic Treatment

**DOI:** 10.3390/ma15062133

**Published:** 2022-03-14

**Authors:** Kanyanat Tawatbundit, Sumrit Mopoung

**Affiliations:** Chemistry Department, Faculty of Science, Naresuan University, Phitsanulok 65000, Thailand; kanyanatt58@nu.ac.th

**Keywords:** hydrothermal, activated carbon, aquaponic treatment, sugarcane leaf

## Abstract

The effects of hydrothermal treatment, 0–5% KMnO_4_ content, and 300–400 °C pyrolysis temperature, were studied for activated carbon preparation from sugar cane leaves in comparison with non-hydrothermal treatment. The percent yield of activated carbon prepared by the hydrothermal method (20.33–36.23%) was higher than that prepared by the non-hydrothermal method (16.40–36.50%) and was higher with conditions employing the same content of KMnO_4_ (22.08–42.14%). The hydrothermal and pyrolysis temperatures have the effect of increasing the carbon content and aromatic nature of the synthesized activated carbons. In addition, KMnO_4_ utilization increased the O/C ratio and the content of C-O, Mn-OH, O-Mn-O, and Mn-O surface functional groups. KMnO_4_ also decreases zeta potential values throughout the pH range of 3 to 11 and the surface area and porosity of the pre-hydrothermal activated carbons. The use of the pre-hydrothermal activated carbon prepared with 3% KMnO_4_ and pyrolyzed at 350 °C as a filter in an aquaponic system could improve the quality of water with pH of 7.2–7.4, DO of 9.6–13.3 mg/L, and the turbidity of 2.35–2.90 NTU. It could also reduce the content of ammonia, nitrite, and phosphate with relative removal rates of 86.84%, 73.17%, and 53.33%, respectively. These results promoted a good growth of catfish and red oak lettuce.

## 1. Introduction

Hydrothermal carbonization is a chemical process, which can convert an organic precursor to biochar in aqueous solution under an oxygen-free atmosphere with temperatures between 150 °C and 270 °C and high pressures between 10 bar and 55 bar [1,2]. It is a low energy demand and environmentally friendly procedure, which can improve the chemical and physical characteristics of the charcoal and result in a high amount of porous structures and surface functional groups [1,3]. Importantly, the process uses water for the treatment, which is economical, non-toxic, and environmentally friendly [4]. The factors that greatly affect the efficiency of the hydrothermal process are the reaction temperature, reaction time, biomass to water ratio, and pressure [5]. Various reactions such as hydrolysis, decarboxylation, dehydration, condensation, aromatization, and polymerization occur during the hydrothermal process [2]. The process starts with hydrolysis and fragmentation of the organic compounds followed by polymerization and condensation reactions to form colloidal coal particles and dewaterability of the char product [1]. During the hydrothermal process, subcritical water is produced, which exhibits high ionic strength in a homogeneous reaction system. This subcritical water can improve the oxygen-containing surface functional groups of hydrocarbons, which enhances the activity of the activating agent and thus yields improved porosity in the carbon during activation [6]. The process also allows stabilizing the macrocellular structure of biomass before pyrolysis, resulting in higher mesoporosity [2]. However, the carbon atoms inside the hydrochar are still hindered by its functional groups or its attachment to other compounds [7]. Thus, after hydrothermal pretreatment of the wet materials, the hydrochar product must be carbonized and activated to obtain true carbon materials and activated carbons with increased carbon content and improved the textural properties [8]. Tests of the application of the hydrothermal pretreated activated carbon have shown that it could exhibit enhanced adsorption of methylene blue by electrostatic interactions and hydrogen bonding interactions of surface oxygen-containing functional groups on the carbon surface [6]. The activated carbons were also used for water treatment and soil amendment, as a solid fuel, and for biomedical purposes [4,5]. In general, wet materials such as sewage sludge [1], organic matter [9], and wet biomass [4] with high water content are suitable for hydrothermal treatment. Especially, the lignocellulosic materials, such as plants and their derived residues and wastes, have been pretreated by the hydrothermal process for hydrochar, bio-oil, and synthesis gas production. This is because organic materials respond well in subcritical water conditions, which are maintained during the hydrothermal process [4]. In addition, the hydrothermal treatment is also performed for the purposes of ash and nitrogen removal, hemi-cellulose decomposition, reduction of H/C and O/C ratios of biomass, and biological sterilization [5,8,9]. Corn stigmata treated with CO_2_ [2], acacia [5], cassava and tapioca flour treated with KOH [7], agave americana fibers and mimosa tannin [8], and rice husk treated with NaOH [9], have been used for activated carbon preparation by hydrothermal pretreatment. Sugarcane leaves are one of the waste products of sugarcane farming, produced with the rate of 6–8 tonnes of dry weight per hectare. It is lignocellulosic biomass waste, which consists of solid and polymeric organic compounds as well as silica. Sugarcane leaves consist of cellulose (31–45%), hemicelluloses (20–30%), and lignin (12–31%) [10]. It has been used for the fabrication of mesoporous materials and silica gel preparation [11]. It was also used for the reinforcement of polymer composites [12]. In addition, sugar cane leaves have been used to produce bioproducts and biomaterials such as ethanol, xylitol, biogas, enzymes, and oligosaccharides after pretreatment by physical, physicochemical, chemical, and biological processes [10]. In this research, the activated carbon from sugarcane leaves was used for water treatment in an aquaponics system. Aquaponics is an integrated system of aquaculture and hydroponics with closed-loop recycling of fresh water within the system between fish and plants [13]. The water in the aquaculture system is circulated to the hydroponics system. During water circulation in the hydroponics system, the nutrients in the water, which are the waste materials from the fish, are absorbed by the plants. The treated water is then recycled back to the aquaculture system [14]. This system could thus reduce the use of land [15]. It is also less energy-intensive, is environmentally friendly, and requires less water consumption with a minimum requirement of chemicals or fertilizers [16]. In addition, it also qualifies as an organic farming method for vegetables and fish production, utilizing sustainable food production technology [17]. However, this system is still more popular as a hobby rather than for commercial- scale production [18]. The problems with this system are turbidity, a high content of toxic nitrogen compounds (ammonia, and nitrite), and low dissolved oxygen in water. To solve these problems, factors such as stocking density, cultivation media, ratios of plants to fish, and water recirculation rates [19] were investigated. Filter suction devices are also used for water treatment in aquaponics [20]. Biological treatments (nitrifying biofilters) and solid capture via mechanical means were used for wastewater treatment [21]. According to previous reports, oyster shell [22], biochar [14], sand, rockwool, glass wool, anthracite, pumice, calcined clay, crushed brick, polyethylene beads [19], rice biofilter [23], and vetiver [24] have been used for solid capture in the water treatment of aquaponics.

In this research, the effects of hydrothermal pretreatment and KMnO_4_ addition used for activated carbon preparation from sugarcane leaves with a high content of functional surface groups were studied to achieve a decrease in the activation temperature. The final activated carbon product with a high content of surface functional groups, percent yield and good stability, which were prepared by the lowest of activation temperatures, was used for the water treatment of aquaponics.

## 2. Materials and Methods

The hydrothermal modification and KMnO_4_ addition were used for the pretreatment of sugarcane leaves at 120 °C, 15 bar, and 0–5% KMnO_4_ concentrations carried out for 6 h with the hope that the constituents in sugarcane leaves will be more easily disintegrated. The products of the hydrothermal pretreatment were activated at 300–400 °C without the addition of activating reagents, to find the lowest temperature for activated carbon production, resulting in products with a high performance in terms of chemical and physical properties and a higher mass yield. The activated carbon was applied for water treatment in an aquaponics system. The water parameters monitored in this study were pH, NH_3_, NO_2_^−^, DO, orthophosphate and turbidity removal efficiency, and growth of catfish and red oak lettuce.

### 2.1. The Production of Sugarcane Leaf-Activated Carbon

Sugarcane leaves were collected from Kamphaengphet, a province in northern Thailand. The precursor materials (10 g) were washed with distilled water, sun-dried, and cut by pruning shears to a particle size in the range of 10–20 mm. The precursor materials were weighted and placed in a container and then potassium permanganate solution with concentration of 0%, 1%, 3%, or 5%wt in distilled water was added (10 mL). The samples were immersed in the container for 24 h. After 24 h, the sample container was placed into a stainless-steel autoclave and treated at 120 °C and 15 bar with a reaction time of 6 h. The autoclave was then cooled down to room temperature (the samples produced from this process will be called hydrochar in further discussion). For samples made without hydrothermal pretreatment, the precursor materials were dried at 100 °C after being immersed in 0%, 1%, 3%, or 5%wt KMnO_4_ in distilled water. The hydrochar samples and non-hydrothermal pretreatment samples were then placed into a crucible with a lid for the heating step. The pyrolysis was conducted at 300 °C, 350 °C, or 400 °C and the pyrolysis temperature was maintained constant for 6 h. The pyrolysis temperature for the preparation of activated carbon from biomass should not exceed 400 °C, a temperature at which hemicellulose, cellulose, and lignin all decompose [25].

### 2.2. Characteristics of Sugarcane Leaf and Sugarcane Leaf Activated Carbon

The sugarcane leaf and sugarcane leaf activated carbon were characterized by Fourier transform infrared spectrometer (FT-IR, Spectrum GX, Perkin Elmer Frontier, Richmond, Llantrisant, UK: to classify the organic, inorganic, and chemical bonds or functional groups), scanning electron microscopy and energy dispersive X-ray spectrometer (SEM, a LEO 1455 VP Electron Microscopy, Oxford instruments, Oxon, Cambridge, UK: and EDS, Edax LED1455P, AMETEK, San Luis Obispo, CA, USA: To study the surface characteristics, size, shape, and as an analytical technique used for the elemental analysis or chemical characterization), X-ray diffraction (XRD, PW 3040/60 X’Pert PRO Console, Philips, Bruker D2 PHASER, Billerica, MA, USA; To analyze and identify the type of compounds and crystal structure of the obtained activated carbon), surface area and porosity analyzer (BET, Micromeritics TriStar II3020, Bavaria, Germany; to analyze surface area characteristics, for example, pore size diameter, surface area, pore volume), and Zeta potential analyzer (Malvern zeta sizer nano -ZS Almelo, Netherlands; to investigate the potential on the surface of particles).

### 2.3. Using the Sugarcane Leaf Activated Carbon for Aquaponic Treatment

Four systems containing fishponds and PVC tube hydroponic systems (2 m × 2 m × 0.7 m) were built. Growth of catfish (50 fish per cubic meter) and red oak lettuce (90 plant per system) was observed and the quality of water was analyzed. A comparison between 4 sets of experiments (Figure 1) was carried out.

Aquaculture system (pond 1) without a hydroponics system for control experiment (Figure 1a)Hydroponic system without aquaculture system (Figure 1b)Aquaponics system (pond 2 and hydroponic system) without activated carbon filter (Figure 1c)Aquaponics system (pond 3 and hydroponic system) with 3 kg activated carbon filter (Figure 1d)

For the aquaponics system, 3 kg of activated carbon was filled in a 50 L plastic tank, which was placed between the hydroponic system and the aquaculture system (Figure 1d). The fish were fed at a rate of 5% by weight at 8:00 a.m. Analysis of the 4 water systems was made through an observation of the growth of catfish (weight of the body) every 7 days for 4 weeks and red oak lettuce by randomly selecting 10% of the sample set at week 4. Dissolved oxygen (DO), pH, and turbidity of water were measured every 7 days for 4 weeks. The content of ammonia, nitrite, and orthophosphate were measured at week 4.

### 2.4. Water Analysis

Water samples were collected weekly from each pond within 20 cm of the water surface at about 09:00 AM by plastic bottles. Turbidity, by turbidimeter (Jenway 6035, Jenway, Mortdale, Australia), and pH (Mettler Toledo) of water were measured. For NH_3_, NO_2_^−^, PO_4_^3−^, and DO analysis, the water samples were filtered through Whatman No.42 paper. NH_3_ was determined by indophenol reaction, nitrite by the diazo-azo colorimetric method, and orthophosphate by an ascorbic acid method with the resulting solutions being measured by spectrophotometer (double beam, Jasco V650) at 422 nm, 540 nm, and 880 nm, respectively. Dissolved oxygen (DO) was also measured by titration with sodium thiosulphate [26]. The percent relative removal rate was used for evaluating the removal efficiency of NH_3_, NO_2_^−^, and orthophosphate from water.
Relative removal rate (%) = [(C_0_ − C_t_)/C_0_] × 100%
where C_0_ refers to the concentration of the control (pond 1), and C_t_ refers to the concentration of each treatment (pond 3 or 4).

## 3. Results and Discussion

The aim of this study was to obtain activated carbon with sufficient chemical and physical stability for wastewater treatment applications. The beneficial properties of the final activated carbon material are its high micropore volume, surface functional groups, and its percent yield.

### 3.1. Percent Yield

Figure 2 shows that the percent yield of all activated carbons prepared from sugar cane leaf decreased as the activation temperature increased from 300 °C to 400 °C. This shows that the samples have undergone more extensive thermal degradation at higher activation temperatures. Furthermore, percent yields of samples prepared with hydrothermal pretreatment are higher than for samples prepared without hydrothermal pretreatment and activated at the same temperature. This is because the pre-hydrothermal process can decompose the hemicellulose in the sugar cane leaves, which causes the aromatisation reaction of pre-hydrothermal samples to begin to take place. This phenomenon makes the pre-hydrothermal samples stable to thermal activation [2,3]. Moreover, the percent yields of pyrolyzed hydrothermal samples also increase with the increasing amount of KMnO_4_ from 1% to 5%. In this case, the surface areas of pre-hydrothermal samples formed after hemicellulose degradation increase with the increasing amount of KMnO_4_. The higher surface area of pre-hydrothermal samples can enhance binding of KMnO_4_ to the samples [3]. In addition, the oxides of Mn and K, which are formed by the decomposition of potassium permanganate, are resistant to thermal degradation at temperatures in the 300 °C–400 °C range, thereby increasing the percent yield of pre-hydrothermal samples as compared to samples made in the absence of potassium permanganate.

### 3.2. Elemental Composition from EDS

For samples made at the same conditions, the carbon content of all pyrolyzed products increases with the increasing activation temperature, while the amount of oxygen shows the opposite trend (Table 1). This phenomenon indicates that the volatile matters were degraded to a greater extent at higher activation temperatures. Considering the effect of the hydrothermal pretreatment, it was found that the carbon content of the pyrolyzed pre-hydrothermal products was higher than that for the without-hydrothermal ones. This is because hemicellulose, which has a high O and H content of pre-hydrothermal samples, decomposes during the hydrothermal exposure [3] at 120 °C and 15 bar for 6 h. These results indicate that the O/C ratios of the pyrolyzed products tend to decrease with the increasing activation temperature as a result of the non-hydrothermal treatment and 0% KMnO_4_ hydrothermal pre-treatment. However, the O/C ratio of pyrolyzed products tends to increase in response to the addition of potassium permanganate and is increased even further when the amount of potassium permanganate used is increased. This is due to the oxidation of the samples with potassium permanganate, resulting in a reduction in carbon content and an increase in oxygen content. Content of other elements in the pyrolyzed products tended to slightly increase with the increase in the activation temperature. In addition, the content of Mn and K increased with the increasing amount of added potassium permanganate. While some of these elements are dissolved and washed away by acetic acid, which forms during the hydrothermal process, Ca and Si resist removal during the hydrothermal treatment [9]. In addition, these elements constitute inorganic matter, which is stable at high temperatures; therefore, it remains in the pyrolyzed products after activation.

### 3.3. FTIR Spectrum of Pyrolyzed Products

The peaks in the FTIR spectra of the pre-hydrothermal samples and the samples without hydrothermal treatment at 3200–3400 cm^−1^ (O-H stretching) and about 2900 cm^−1^ (C-H stretching) disappeared after activation at 300 °C or higher temperatures (Figure 3a–o), which is due to the pyrolytic degradation of hydrogen. Furthermore, when considering the effect of hydrothermal treatment on the surface functional groups of activated carbon, the results show that the peaks in the FTIR spectrum at 1712 cm^−1^ (C=O stretching of hemicellulose), 1384 cm^−1^ (-C-O-H) [27], and 622 cm^−1^ (C-H and/or aryl C-O groups, 1) of samples without hydrothermal treatment (Figure 3a–c) are more intense than those for the pre-hydrothermal samples (Figure 3d–f) for 0% KMnO_4_ at the equivalent activation temperature. It is possible that the hydrothermal effect causes the degradation of the C=O group and other functional groups of hemicellulose at 120 °C and 15 bar. Conversely, the peak at 1604 cm^−1^ (aromatic C=C) shows the opposite result. It has been suggested that the hydrothermal effect increases the content of aromatics in the pre-hydrothermal samples to a larger extent than in samples without hydrothermal treatment due to dehydration and decarboxylation reactions [1]. Moreover, the 1604 cm^−1^ peak intensity of both samples increases as the activation temperature increases from 300 °C to 400 °C (Figure 3a–o). It has been shown that the content of aromatics in pyrolyzed products increases with the increasing activation temperature, which promotes the condensation [1]. In regards to the effects of pre-hydrothermal treatment and activation temperature, the FTIR peaks in pre-hydrothermal samples at 1712 cm^−1^, 1604 cm^−1^, and 622 cm^−1^ decrease with the increasing activation temperature from 300 °C to 400 °C for samples with 0% KMnO_4_ content (Figure 3d–f). In addition, increasing the content of KMnO_4_ in pre-hydrothermal samples results in FTIR spectra where the intensity of peaks at 1712 cm^−1^, 1604 cm^−1^ (C=C stretching), and 622 cm^−1^ have the tendency to decrease with the increasing content of KMnO_4_, while the peaks at 1094 cm^−1^ (C-O stretching), 786 cm^−1^, and 444 cm^−1^ have the opposite trend for all activation temperatures between 300 °C and 400 °C (Figure 3d–o). Furthermore, the peaks in the FTIR spectrum of the pre-hydrothermal samples at 1712 cm^−1^ disappeared after activation at 350 °C for samples pretreated with 1–5% KMnO_4_ (Figure 3h,i,k,l,n,o). In this case, potassium permanganate was expected to have a greater effect on the degradation of surface functional groups in activation products through more extensive decarboxylation and aliphatic degradation [1]. These reactions were complete for samples activated at 350 °C after pretreatment with 3–5% KMnO_4_, which indicates the stability of the final products. However, potassium permanganate also increased partial oxidation, resulting in an increased content of C-O (peak at 1094 cm^−1^) surface functional groups and residual content of Mn-OH (622 cm^−1^), O-Mn-O, and Mn-O bonds (444 cm^−1^) [28] on the surface of activation products. The peaks at 1604 cm^−1^ and 1384 cm^−1^ are related to aromatic skeletal vibrations of lignin [29]. In addition, the peak at 1094 cm^−1^ is also related to the C-O and C-O-C stretching vibrations, which is characteristic of the anomeric region of cellulose-like structures [3]. These features show that there was still lignin and cellulose left in the pyrolyzed samples. This is because cellulose and lignin completely degrade at pyrolysis temperatures of 450 °C or more [1]. Furthermore, the products also exhibit FTIR vibrations consistent with the presence of functional groups containing Si–O–Si (1094 cm^−1^ and 444 cm^−1^) and Si-O bonds (786 cm^−1^) [30]. The presence of these functional groups in the pyrolyzed products is in good agreement with the elemental composition of the pyrolyzed products, as shown in Table 1. Therefore, the aromatic nature and presence of these functional groups in the pyrolyzed pre-hydrothermal products shows their good stability and potential for adsorption efficiency.

### 3.4. XRD Diffractogram of Pyrolyzed Products

X-ray diffraction patterns of pyrolyzed samples are shown in Figure 4. The bands with peaks at about 26.5° and 43.5° in the XRD patterns of all pyrolyzed samples are assigned to graphite flakes in a disordered layer [31]. It should be pointed out that all samples were converted to charcoal by pyrolysis in the temperature range of 300 °C to 400 °C. However, the products also have differing characteristics as a result of pyrolysis temperature, hydrothermal treatment, and the addition of various amounts of KMnO_4_. For the effect of pyrolysis temperature on the samples without hydrothermal pretreatment (Figure 4a–c), the diffractogram peaks of all pyrolyzed samples without hydrothermal pretreatment show a little peak of cellulose at θ = 15.1° for pyrolysis in the temperature range of 300 °C to 400 °C. This shows that there still was cellulose left in the pyrolyzed products made without hydrothermal treatment. However, this peak decreased with the increasing pyrolysis temperature. This is because cellulose does partially decompose in the temperature range of 240 °C to 350 °C [25]. For hemicellulose, it is expected that it completely degrades in the pyrolysis temperature range of 200 °C to 260 °C [25]. In addition, these pyrolyzed products made without hydrothermal pretreatment were assumed to still contain lignin as lignin decomposes over a range of pyrolysis temperatures from 280 °C to 500 °C [25]. Lignin is amorphous [25] and therefore its XRD peak is not observed. In addition, the diffractogram peaks for products made at pyrolysis temperature of 400 °C also show more pronounced peaks of SiO_2_ (24.6°, 26.5°, 28.4°, 29.9°, and 43.5°) [32], CaO (26.5°, 28.4°, 31.1°, and 36.3°), and K_2_O (26.5°, 28.4°, and 40.8°) [33], due to the decomposition of various volatile substances. The compositions of these substances are derived from the elements, which are present in the original sugar cane leaves. In the same way, pyrolyzed products made with pre-hydrothermal treatment have shown the same trend of diffractogram peaks (Figure 4d–f) to that of the pyrolyzed products made without hydrothermal treatment. One exception is the XRD peak at 15.1° corresponding to cellulose, which has completely disappeared at the pyrolysis temperature of 400 °C (Figure 4f). This suggests that hydrothermal treatment affects the breakdown of cellulose in sugarcane leaves. Likewise, when 1–3% KMnO_4_ was added to the pre-hydrothermal samples (Figure 4g–o), it was found that the XRD diffractograms of these pyrolyzed products exhibit the same effect as the products made with pre-hydrothermal treatment without KMnO_4_ addition in the pyrolysis temperature range of 300 °C to 400 °C. This suggests that the addition of KMnO_4_ had no effect on cellulose degradation in sugarcane leaves.

### 3.5. Zeta Potential of Pyrolyzed 0–1% KMnO_4_ Pre-Hydrothermal Products

Zeta potential is used to characterize the surface electric charge of an absorbent in solution [31]. It is one parameter that affects the affinity between the adsorbent material and the adsorbate [34], which could give a preliminary estimate of the adsorption capacity of the adsorbate for adsorbent materials [35]. Figure 5 shows only the pH dependence of the zeta potential for 0% and 1% KMnO_4_ pre-hydrothermal activated carbon prepared at 350 °C. This experiment was carried out only to determine the effect of potassium permanganate on activated carbon that will be used for further water treatment. The results show that the zeta potential values of both activated carbons are negative throughout the pH range from 3 to 11, and are thus able to capture positively charged adsorbates with electrostatic force [31]. This is due to the electrons from aromatic rings on the surface of the activated carbon being involved in *π*–*π* interactions [34]. Furthermore, adding 1% KMnO_4_ to pre-hydrothermal sample results in a lowering of the zeta potential value of activated carbon (Figure 3b). This is caused by the aromatic ring on the surface of the activated carbon being oxidized with potassium permanganate, which results in the destruction of the aromatic rings. In addition, C-O-H surface functional groups of hemicellulose were partially oxidized to carbonyl groups. This results in a large amount of electric charge accumulation on the surface of the KMnO_4_-treated pre-hydrothermal activated carbon and the decrease of the zeta potential values [36]. Furthermore, the zeta potential of activated carbons decreases with the increase of the solution pH over a range of pH 3.0−7.0 and then increases to a maximum value at pH 9. After pH 9, the zeta potentials of the activated carbons decrease again until pH 11, which is once again due to *π*–*π* interactions within the second layer, formed after the formation of the first layer on the surface of the activated carbon [34].

### 3.6. Morphologies of Pyrolyzed Products

The hydrothermal treatment effect is evident in that it can cause a more extensive decay of the sugar cane leaves. It is evident that the pyrolyzed product made without hydrothermal treatment from sugar cane leaves and prepared at 350 °C has a cellular structure with interconnected porous lattice structures and undamaged cell walls (Figure 6a). The pyrolysis products of the pre-hydrothermal sugar cane leaves show damaged cell walls even at the pyrolysis temperature of 300 °C (Figure 6b). The cell walls of hydrothermal sugarcane leaves are further decomposed and form particles on the surface of the pyrolyzed product with an increasing pyrolysis temperature from 350 °C to 400 °C (Figure 6c,d). Especially at the pyrolysis temperature of 400 °C, the surface of the pyrolyzed product has been disintegrated to a large extent (Figure 6d). When adding potassium permanganate, small particles present on the pyrolyzed products of the pre-hydrothermal sugar leaf are destroyed to an even greater extent and few remain as compared with the pyrolyzed products made without hydrothermal treatment (Figure 6c) and with pre-hydrothermal treatment (Figure 6e) prepared at the same pyrolysis temperature (only products made at pyrolysis temperature of 350 °C are shown). Likewise, the addition of potassium permanganate has a significant effect on the breakdown of surface particles on pyrolyzed products of pre-hydrothermal samples with more surface particles being decomposed and almost completely gone after adding potassium permanganate in the concentration range from 1% to 5% (Figure 6e–g). The surface structure of the pyrolyzed pre-hydrothermal product begins to break at 3% KMnO_4_, and the holes in the sugar cane leaf structure become apparent. Moreover, the surface structures of pyrolyzed pre-hydrothermal samples were very fractured after preparation at a pyrolysis temperature of 400 °C in comparison with pyrolyzed and 5% KMnO_4_-treated pre-hydrothermal products prepared at 350 °C (Figure 6g) and pyrolyzed and 5% KMnO_4_-treated pre-hydrothermal products prepared at 400 °C (Figure 6h). This is because the decomposition of volatile organic compounds of sugar cane leaves, which is due to hydrothermal treatment, KMnO_4_, and pyrosis temperature.

### 3.7. Surface Characteristic of Sugarcane Leaf Activated Carbon

The surface areas of pyrolyzed products made with hydrothermal treatment using 0% KMnO_4_ are higher than those of products made without hydrothermal treatment. In addition, the surface area increases with the increasing activation temperature from 300 °C to 400 °C for both types of pyrolyzed products (Table 2). The surface areas and micropore volumes of pyrolyzed products made with hydrothermal treatment using 0% KMnO_4_ are higher than for samples made without hydrothermal treatment. This is because hydrothermal treatment can remove hemicellulose and some cellulose to generate pores on the pyrolyzed products [3]. In addition, the content of pores increases with the increasing activation temperature from 300 °C to 400 °C for both types of pyrolyzed products (Table 2). This is due to the effect of higher pyrolysis temperatures causing more extensive breakdown of cellulose and lignin present in the initial samples. The addition of KMnO_4_ during the hydrothermal pretreatment results in an increased surface area and micropore volume in comparison to samples made without KMnO_4_ at all pyrolysis temperatures. In addition, surface area and micropore volume also increase with an increasing temperature and potassium permanganate content. This is due to the ability of potassium permanganate to oxidize components of sugar cane leaves. However, the surface area and micropore volume of pyrolyzed 5% KMnO_4_ pre-hydrothermal products decreased below those of pyrolyzed 3% KMnO_4_ pre-hydrothermal products at all temperatures. This is due to the presence of excess potassium permanganate, which covers the surface of the pyrolyzed products. Furthermore, the micropore volume of products pyrolyzed at 300 °C made with 3% KMnO_4_ pre-hydrothermal treatment is higher in comparison to the products pyrolyzed at 400 °C. However, the surface area of products pyrolyzed at 300 °C made with 3% KMnO_4_ pre-hydrothermal treatment is close to products pyrolyzed at 400 °C made with 1% KMnO_4_ pre-hydrothermal treatment.

### 3.8. The Results of Water Analysis

The activated carbon made by pyrolysis at 350 °C and with 3% KMnO_4_ pre-hydrothermal treatment was used as a filter in an aquaponic system. This material has a high micropore volume, a low content of C-H surface functional groups, and a stable physical and chemical structure, although some of its properties are inferior to products made by pyrolysis at 400 °C. This material was chosen to reduce the energy consumption in the preparation of activated carbon.

#### 3.8.1. pH of Water

The pH of water in pond 1, or aquaculture (control treatment, without connection with plant system and activated carbon), is weakly acidic (Figure 7a), while the pH of water in hydroponic (only plant system), is weakly basic (Figure 7b). After connecting the fishpond to the plant system (pond 2 or aquaponic pond, Figure 7c), the pH of the water in the resulting pond 2 has changed from weakly basic (in the first week) to weakly acidic (2–4 weeks) during the course of the experiment. However, the pH of water in pond 3 (aquaponic pond with activated carbon) remained weakly basic for the whole time (Figure 7d). These phenomena are the result of changing the amount of CO_2_ and O_2_ present in the water from the processes of respiration and photosynthesis. In the case of pond 1, the respiration of the fish produces CO_2_, which can dissolve in water and change into carbonic acid. However, the process of photosynthesis of phytoplankton or microalgae reduces the amount of protons (H^+^) and CO_2_ in the water and increases the oxygen content under light conditions, resulting in a gradual increase in the water pH [37]. Nonetheless, it is not sufficient to neutralize the acidity of the water in the conditions of aquaculture alone. Therefore, the water pH increases when the fish farming system is connected to the crop system (pond 2). This is attributed to the absorption of nutrients, metal ions, and other salts and photosynthetic activities (H^+^ uptake) by roots of the plants [38]. Moreover, the water pH increased further in response to the addition of activated carbon to the aquaponic system (pond 3). This is because the activated carbon, which has a negatively charged surface, can adsorb positively charged protons. The water pH of all of the ponds is in the optimal range for the growth of phytoplankton [39].

#### 3.8.2. Dissolved Oxygen in Water

Dissolved oxygen (DO) in water is the basic requirement for survival for aquatic animals [40]. It can be seen that the DO of water in pond 1 (Figure 8a) is low in comparison to other ponds (pond 2–3). The DO value of water is higher in the aquaponic system (pond 2) (Figure 8c). This increase in DO content is even higher for the pond with the activated carbon filter system (pond 3) (Figure 8d). This is due to the effects of both photosynthesis and respiration. Oxygen is used in the respiration of fish, plant roots, phytoplankton, microbes, and during the nitrification of NH_4_^+^ [39,41]. On the other hand, the photosynthesis performed by microalgae increases the dissolved oxygen content [38]. In addition, the amount of dissolved oxygen can also be increased by gas exchange between atmosphere and water in the circulation system or by a mechanical aeration of the pond. Therefore, DO increases in the aquaponic system (pond 2, Figure 8c), which is due to water circulation and aerenchyma of plant root zones in the pond [39]. Furthermore, DO is further increased after inclusion of activated carbon (pond 3, Figure 8d), which can increase water pH and oxidation of manganese oxide present in activated carbon. It can be seen that the amount of DO in the water for all ponds increases with an increasing of the time of raising the fish. This is due to the presence of an induction phase for microorganism growth in water during which minimal increases in cell density occurs, or when the physiological adaptation of the cell metabolism to growth in the first week takes place [42]. In addition, this also resulted in the presence of more plant roots over the experimental time, which helps to add oxygen to the water. Different rates of oxygen exchange from aerial tissue into the root zone primarily contribute to the differences in DO levels among the plant cultures [38]. This is also related to having more plant roots over time, which helps to add more oxygen to the water.

Turbidity of water is an important indicator of the amount of suspended materials and microorganisms in water, which can have many negative effects on aquatic life [38]. Turbidity can prevent light penetration through the water and can consequently decrease the photosynthetic activity of phytoplankton [43]. It can be seen that the turbidity of water is quite high in pond 1 (Figure 9a) and is low in the hydroponic system (Figure 9b). Furthermore, it is lower in the aquaponic system (pond 2, Figure 9c) and the aquaponic system supplemented with activated carbon (pond 3, Figure 9d). In the case of pond 1, metabolic wastes from fish after feeding are not filtered and adsorbed by any material. Therefore, the turbidity value is quite high and increases over time. For aquaponic culture (pond 2), the turbidity value is highly reduced by the adsorption and filtration of suspended organic matters through plant roots during water circulation and this decreases over time. This is because plant roots grow more over time. Furthermore, the turbidity is more extensively reduced for the aquaponic cultures supplemented with activated carbon by filtration and adsorption through plant roots and activated carbon.

The removal and relative removal rate of NH_3_-N (expressed in the form of total ammonia, NH_3_ + NH_4_^+^), NO_2_^—^N, and orthophosphate from water in ponds 1–4 at week 4 are shown in Table 3. The results show that the concentrations of all the species decreased in the aquaponic system (pond 2) and the aquaponic system, supplemented with activated carbon (pond 3), in comparison to control pond 1. In the case of NH_3_-N, the content was lower than the recommended values in production pond management practices, which is below 2 mg/L for all ponds [44]. However, the value was quite high for pond 1 as it is formed as a waste product of protein metabolism. Furthermore, it was oxidized and transformed to nitrate, via nitrite, under aerobic conditions by nitrification through the actions of ammonia-oxidizing and nitrite-oxidizing bacteria [45]. It also accumulates in the reducing environment of the pond bottom and is released continuously into the water [46]. Its value was lowered for the aquaponic system (pond 2) with a 70.39% relative removal rate and with a 86.84% relative removal rate for aquaponic culture supplemented with activated carbon (pond 3), which is the result of having a high DO content. High dissolved oxygen concentration promotes ammonia oxidation by nitrification and decreases the toxicity of NH_3_ to fish. In addition, nitrogen compounds such as NO, N_2_O, and N_2_ [45], which form from nitrification as well as denitrification, were absorbed by plant roots for growth. Another way of NH_3_ removal, which exists for both ionized (NH_4_^+^) and unionized (NH_3_) ammonia [40], is the removal by NH_4_^+^ adsorption of KMnO_4_-treated pre-hydrothermal activated carbon. As a result, more ammonia was removed in pond 3. Nitrite, which forms by oxidation of NH_3_ and is subsequently transformed into NO_3_^−^ by nitrification bacteria, is removed in the same way as ammonia. However, it is still higher than the appropriate content in aquaculture water, which should be less than 0.1 mg/L [35], except in the hydroponic system. The relative removal rates are 41.46% and 73.17% for pond 2 and pond 3, respectively. It can be seen that nitrite content of pond 3 is close to the appropriate level, which is the result of high DO values. In the case of orthophosphate (PO_4_^3−^), it is also reduced with relative removal rates of 44.44% and 53.33% for pond 2 and pond 3, respectively. This is mainly the result of absorption by plant roots for plant growth.

### 3.9. Growth of Fish and Plants

The growth of fish weight in ponds 1, 2, and 3 are presented in Figure 10. It can be seen that fish growth is higher for the aquaponic system (pond 2, Figure 10b) and the aquaponic system supplemented with activated carbon (pond 3, Figure 10c) as compared to the aquatic culture (pond 1, Figure 10a) only in the first week. The fish growth in pond 2 and pond 3 continued to be higher than that of pond 1 until the 4th week. Moreover, the growth of fish in pond 3 is higher than in pond 2 over the course of 4 weeks. These results indicate that pH, DO content, turbidity, NH_3_ content level, and NO_2_^−^ content levels in pond 2 and pond 3 are suitable for improved fish growth. The plant growth data for all crop panels collected in the 4th week are shown in Table 4. It can be seen that among all of the growth parameters, plant crop tube panel 2 (aquaponic) and plant crop tube panel 3 (aquaponic system supplemented with activated carbon) tend to be slightly higher than crops in tube panel 1 (hydroponic system). A higher plant growth also resulted in a higher removal efficiency of NH_3_, NO_2_^−^, and other species. This is especially the case with the growth of plant roots, which has a great effect on the adsorption of various chemicals and DO, since the plant roots have an important function in reducing nitrogen and increasing DO. The roots form habitats for microbes, which cause nitrification/denitrification and also nutrient absorption in the hydroponic plant [47]. Thus, it can be expected that the level of ammonia, nitrite, and turbidity would be substantially lower in an efficient aquaponic system and an aquaponic system supplemented with activated carbon due to the continual uptake or adsorption and in filtering by the plant roots and activated carbon.

## 4. Conclusions

Hydrothermal treatment, potassium permanganate content, and pyrolysis temperature all had a significant effect on the properties of activated carbon prepared from sugar cane leaves. The percent yields of all activated carbons decrease with the increase in the activation temperature from 300 °C to 400 °C. The percent yield of hydrothermal-activated carbon was higher than non-hydrothermal-activated carbon prepared using the same conditions and increased with the potassium permanganate concentration used. Likewise, the carbon content of activated carbon increases with hydrothermal processing and increasing pyrolysis temperature, while the oxygen content exhibits the opposite trend. However, the O/C ratio of pyrolyzed activated carbons increased as the potassium permanganate content increased. The surface functional groups of activated carbon have been eliminated as a result of the effects of hydrothermal pretreatment and increasing pyrolysis temperature, while the content of aromatics has increased. On the other hand, the addition of potassium permanganate increased the presence of C-O, Mn-OH, O-Mn-O, and Mn-O bonds as surface functional groups on pyrolyzed KMnO_4_-treated pre-hydrothermal products. The XRD results also confirm that the pyrolyzed KMnO_4_-treated pre-hydrothermal products were amorphous and contained oxides of K and Mn. In addition, the zeta potential values of pre-hydrothermal activated carbons are negative throughout the pH range from 3 to 11 and are more negative for materials made with added potassium permanganate. Likewise, the addition of potassium permanganate and hydrothermal treatment have a significant effect on the breakdown and degradation of surface particles in pyrolyzed products. These results indicate that the surface area and porosity of the KMnO_4_-treated pre-hydrothermal-activated carbon materials increased. The activated carbon made with 3% KMnO_4_ pre-hydrothermal treatment and pyrolyzed at 350 °C was used as a filter material in an aquaponic system. The results from these experiments have shown that the water was weakly alkaline (pH (7.2–7.4)), had a DO value suitable for fish farming (9.6–13.3 mg/L), and that its turbidity was significantly reduced (reduced from 7.73–8.31 NTU to 2.35–2.90 NTU). It has also been found that pyrolyzed activated carbon made with KMnO_4_ pre-hydrothermal treatment can significantly reduce ammonia (relative removal rate = 86.84%), nitrites (relative removal rate = 73.17%), and phosphates (relative removal rate = 53.33%), as a result of adsorption and filtration on activated carbon and plant roots and that these species are used for plant growth. The water properties found in the aquaponic system supplemented with activated carbon appear to be suitable for fish growth, as the fish showed good growth for 4 weeks. The results of this research indicate that the hydrothermal treatment at the low temperature of 120 °C at 15 bar with 1–3% KMnO_4_ content and pyrolysis at 350 °C are suitable for activated carbon production from sugar cane leaves to be used in aquaponic systems of catfish and red oak. The combined process can decompose surface functional groups of sugarcane leaves more easily during pyrolysis. In addition, the process also results in the formation of aromatics in the pyrolysis products, with a high yield at only 350 °C. Furthermore, KMnO_4_ causes partial oxidation, which induces a greater amorphousness and negative zeta-potential in the activated carbon throughout the pH range from 3 to 11. This in turn increases the affinity between the activated carbon and positively charged adsorbates. These results imply a reduced energy consumption for activated carbon production and improved adsorption properties of the activated carbon.

## Figures and Tables

**Figure 1 materials-15-02133-f001:**
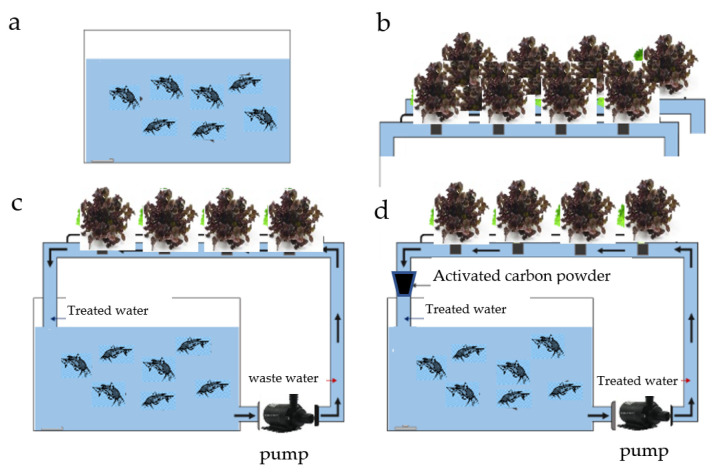
(**a**) aquaculture system; (**b**) hydroponic system; (**c**) aquaponics system; and (**d**) aquaponics system with activated carbon filter.

**Figure 2 materials-15-02133-f002:**
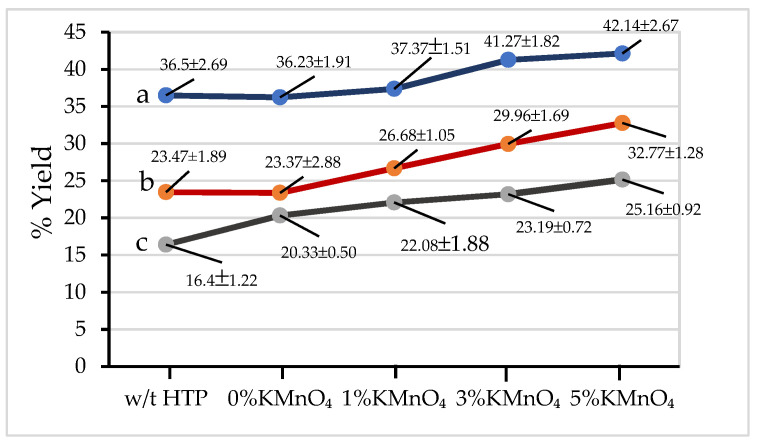
Percent yield of activated carbon materials made with or without hydrothermal pretreatment and with addition of KMnO_4_ of 0–5% by weight at (**a**) 300 °C (**b**) 350 °C and (**c**) 400 °C.

**Figure 3 materials-15-02133-f003:**
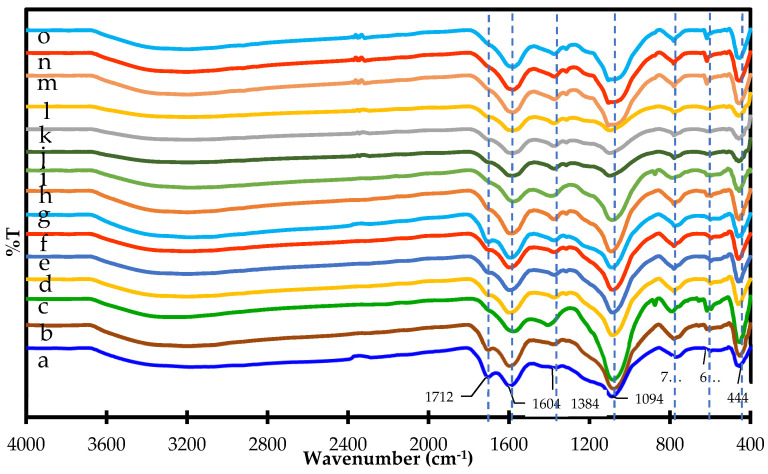
FTIR transmission of (a) activated carbon w/t HTP prepared at 300 °C; (b) activated carbon w/t HTP prepared at 350 °C; (c) activated carbon w/t HTP prepared at 400 °C; (d) activated carbon made with 0% KMnO_4_ HTP and prepared at 300 °C; (e) activated carbon made with 0% KMnO_4_ HTP and prepared at 350 °C; (f) activated carbon made with 0% KMnO_4_ HTP and prepared at 400 °C; (g) activated carbon made with 1% KMnO_4_ HTP and prepared at 300 °C; (h) activated carbon made with 1% KMnO_4_ HTP and prepared at 350 °C; (i) activated carbon made with 1% KMnO_4_ HTP and prepared at 400 °C; (j) activated carbon made with 3% KMnO_4_ HTP and prepared at 300 °C; (k) activated carbon made with 3% KMnO_4_ HTP and prepared at 350 °C; (l) activated carbon made with 3% KMnO_4_ HTP and prepared at 400 °C; (m) activated carbon made with 5% KMnO_4_ HTP and prepared at 300 °C; (n) activated carbon made with 5% KMnO_4_ HTP and prepared at 350 °C; (o) and activated carbon made with 5% KMnO_4_ HTP and prepared at 400 °C.

**Figure 4 materials-15-02133-f004:**
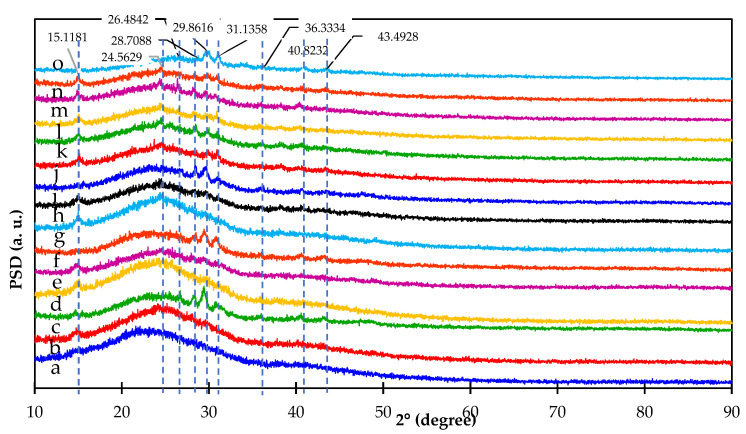
XRD patterns of (a) activated carbon w/t HTP prepared at 300 °C; (b) activated carbon w/t HTP prepared at 350 °C; (c) activated carbon w/t HTP prepared at 400 °C; (d) activated carbon made with 0% KMnO_4_ HTP and prepared at 300 °C; (e) activated carbon made with 0% KMnO_4_ HTP and prepared at 350 °C; (f) activated carbon made with 0% KMnO_4_ HTP and prepared at 400 °C; (g) activated carbon made with 1% KMnO_4_ HTP and prepared at 300 °C; (h) activated carbon made with 1% KMnO_4_ HTP and prepared at 350 °C; (i) activated carbon made with 1% KMnO_4_ HTP and prepared at 400 °C; (j) activated carbon made with 3% KMnO_4_ HTP and prepared at 300 °C; (k) activated carbon made with 3% KMnO_4_ HTP and prepared at 350 °C; (l) activated carbon made with 3% KMnO_4_ HTP and prepared at 400 °C; (m) activated carbon made with 5% KMnO_4_ HTP and prepared at 300 °C; (n) activated carbon made with 5% KMnO_4_ HTP and prepared at 350 °C; (o) and activated carbon made with 5% KMnO_4_ HTP and prepared at 400 °C.

**Figure 5 materials-15-02133-f005:**
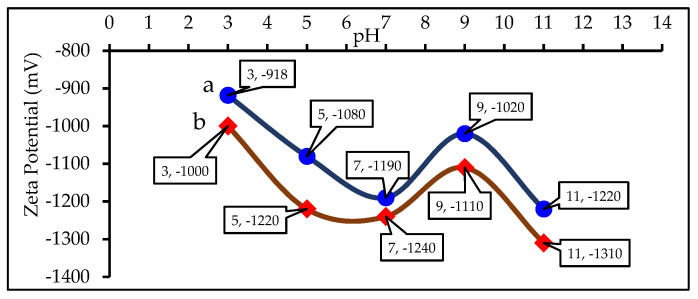
Zeta Potential of (a) activated carbon treated with 0% KMnO_4_ HTP and prepared at 350 °C and (b) activated carbon treated with 1% KMnO_4_ HTP and prepared at 350 °C.

**Figure 6 materials-15-02133-f006:**
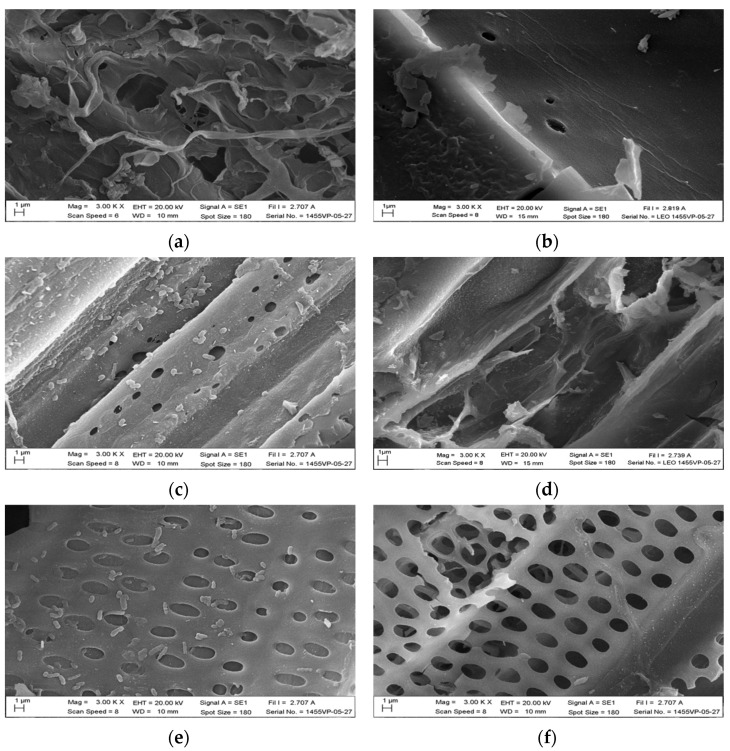
SEM morphologies acquired with a 3000 magnification of (**a**) activated carbon product pyrolyzed without hydrothermal at 350 °C; (**b**) activated carbon product made with 0% KMnO_4_ pre-hydrothermal treatment and pyrolyzed at 300 °C; (**c**) activated carbon product made with 0% KMnO_4_ pre-hydrothermal treatment and pyrolyzed at 350 °C; (**d**) activated carbon product made with 0% KMnO_4_ pre-hydrothermal treatment and pyrolyzed at 400 °C; (**e**) activated carbon product made with 1% KMnO_4_ pre-hydrothermal treatment and pyrolyzed at 350 °C; (**f**) activated carbon product made with 3% KMnO_4_ treatment and pyrolyzed at 350 °C; (**g**) activated carbon product made with 5% KMnO_4_ pre-hydrothermal treatment and pyrolyzed at 350 °C; (**h**) and activated carbon product made with 5% KMnO_4_ pre-hydrothermal treatment and pyrolyzed at 400 °C.

**Figure 7 materials-15-02133-f007:**
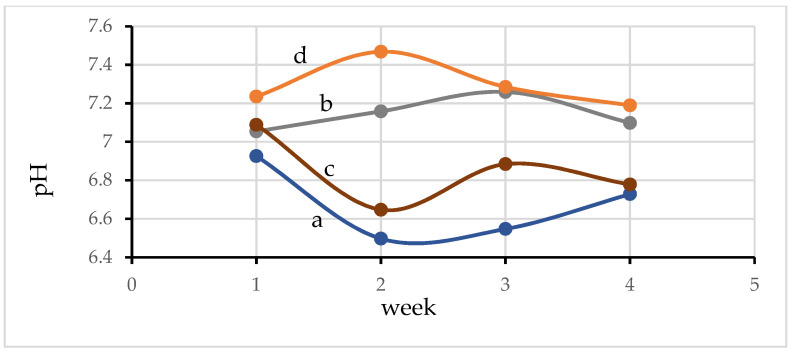
pH values of water of (a) pond 1; (b) the hydroponic system without an aquaculture pond; (c) pond 2; (d) and pond 3.

**Figure 8 materials-15-02133-f008:**
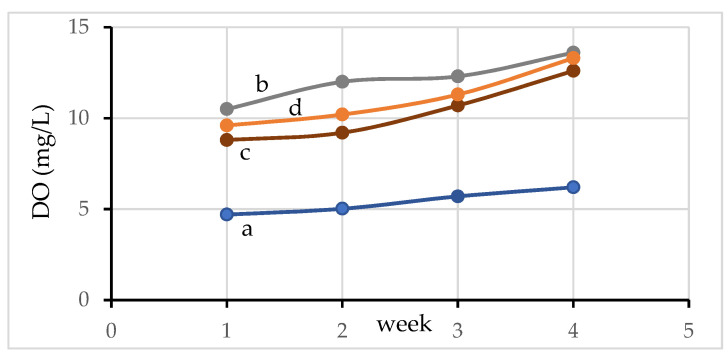
DO values of the water of (a) pond 1 (b) hydroponic system without an aquaculture pond (c); pond 2; and (d) pond 3.

**Figure 9 materials-15-02133-f009:**
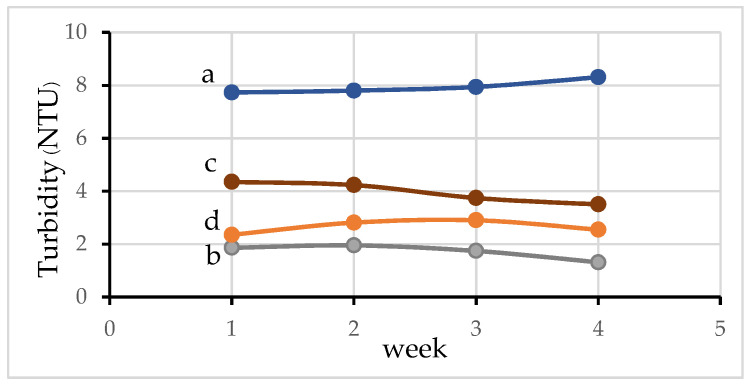
Turbidity values of the water in (a) pond 1; (b) the hydroponic system without aquaculture pond; (c) pond 2; (d) and pond 3.

**Figure 10 materials-15-02133-f010:**
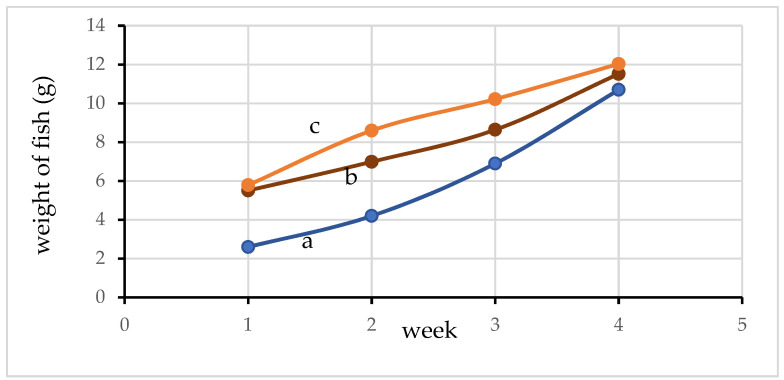
The fish growth observed in (a) pond 1; (b) pond 2; (c) and pond 3.

**Table 1 materials-15-02133-t001:** Elemental composition of pyrolyzed products from EDS.

Temp. (°C)	SLACs * Sample with	Elements Composition (%wt.)						
		C	O	O/C Ratio	Mn	Si	K	Ca
300	w/t HTP	60.87	35.85	0.59	0.38	1.32	0.57	0.81
	0%KMnO_4_^−^ HTP	69.93	21.66	0.31	1.01	4.06	1.38	0.96
	1%KMnO_4_^−^ HTP	62.83	27.44	0.39	1.05	4.87	2.46	1.39
	3%KMnO_4_^−^ HTP	57.45	30.67	0.53	1.26	4.87	4.23	1.45
	5%KMnO_4_^−^ HTP	50.55	29.86	0.59	5.64	7.04	4.70	2.18
350	w/t HTP	65.52	30.85	0.47	0.67	1.83	0.32	1.01
	0%KMnO_4_^−^ HTP	70.69	26.10	0.37	0.45	0.58	0.59	1.27
	1%KMnO_4_^−^ HTP	64.70	27.35	0.42	0.85	3.58	2.57	0.94
	3%KMnO_4_^−^ HTP	61.09	28.62	0.42	1.16	5.50	2.81	1.68
	5%KMnO_4_^−^ HTP	54.15	32.23	0.60	1.90	6.53	3.27	1.93
400	w/t HTP	68.46	28.28	0.41	0.45	1.45	0.97	1.22
	0%KMnO_4_^−^ HTP	75.05	20.17	0.27	0.60	1.67	1.58	1.23
	1%KMnO_4_^−^ HTP	71.68	21.29	0.30	0.66	2.43	2.54	1.40
	3%KMnO_4_^−^ HTP	70.88	21.83	0.31	0.82	3.65	2.60	0.21
	5%KMnO_4_^−^ HTP	59.83	26.44	0.44	1.21	7.74	3.86	0.60

* SLACs is sugar cane leaves; w/t HTP signifies samples without hydrothermal pre-treatment and KMnO_4_ treatment.

**Table 2 materials-15-02133-t002:** Surface area and porosity of sugarcane leaf activated carbon determined by BET.

Temp. (°C)	SLACs * Sample with	BET Surface Area (m^2^/g)	Surface Area of Pores between 17 Å and 3000 Å (m^2^/g)	Micropore Volume (cm^3^/g)	Volume of Pores between 17Å and 3000 Å (cm^3^/g)
300	w/t HTP	0.9289	0.1997	0.000698	0.000089
	0%KMnO_4_^−^ HTP	2.9560	0.9345	0.006240	0.000921
	1%KMnO_4_^−^ HTP	5.9670	3.2251	0.003900	0.004285
	3%KMnO_4_^−^ HTP	13.5882	9.2956	0.006700	0.013424
	5%KMnO_4_	7.5572	5.8829	0.002610	0.006508
350	w/t HTP	10.8815	8.8915	0.002647	0.003702
	0%KMnO_4_^−^ HTP	12.1574	11.7152	0.007006	0.011131
	1%KMnO_4_^−^ HTP	15.1145	13.6461	0.002525	0.071490
	3%KMnO_4_^−^ HTP	30.5653	26.0662	0.004230	0.2744
	5%KMnO_4_^−^ HTP	24.8480	19.5535	0.001575	0.16119
400	w/t HTP	23.7722	18.1505	0.002759	0.18773
	0%KMnO_4_^−^ HTP	24.9388	21.0224	0.005924	0.13548
	1%KMnO_4_^−^ HTP	33.3101	31.7505	0.009794	0.14507
	3%KMnO_4_^−^ HTP	45.0364	42.4550	0.002290	0.11473
	5%KMnO_4_^−^ HTP	34.7107	32.7012	0.002200	0.022678

* SLACs is sugar cane leaves.

**Table 3 materials-15-02133-t003:** The removal and percent relative removal rate of NH_3_, NO_2_^−^, and orthophosphate.

Pond Number	Removal of					
	NH_3_		NO_2_^−^		PO_4_^3−^ mg/L	
	mg/L	Removal Rate %	mg/L	Removal Rate %	mg/L	Removal Rate %
1	1.52	-	0.41	-	0.45	-
Hydroponic	0.04	-	0.02	-	0.02	-
2	0.45	70.39	0.24	41.46	0.25	44.44
3	0.20	86.84	0.11	73.17	0.21	53.33

**Table 4 materials-15-02133-t004:** The growth parameters of red oak in the 4th week.

Crop Tube Panel Number	Stem Height (cm)	Dried Stem Weight (g/stem)	Root Length (cm)	Dried Root Weight (g/stem)	Number of Leaves/Stem	Dried Leaf Weight (g)	Trunk Diameter (cm)
1	15.63	1.12	9.32	0.24	9.2	0.14	16.4
2	16.87	1.26	10.22	0.26	9.3	0.15	16.4
3	16.89	1.47	11.22	0.29	9.3	0.17	17.75

## Data Availability

Data sharing not applicable.

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
