# Peer review of "Activated Carbon Preparation from Sugarcane Leaf via a Low Temperature Hydrothermal Process for Aquaponic Treatment"

_materials, 2022, doi:10.3390/ma15062133_

Round 1

Reviewer 1 Report

The manuscript by Tawatbundit and Mopoung discuss the effects of different preparation conditions for activated carbon from sugar cane leaves. The optimization of the process in terms of yield, surface functional groups and chemical-physical properties of the activated carbon is presented. Moreover, the obtained material was used as an example for a practical application, the growth of catfish and red oak lettuce in an aquaponics system.

Some small errors are presented in the manuscript:

In the abstract please add the word lettuce after red oak and in conclusion change decease with decrease. Other small English errors are present in the text, please give the manuscript a careful reading.

When referred to the hydrothermal pretreatment, the authors abbreviate HPT in the tables but HTP in the figures. Please keep the same notation.

In the Materials and methods, it is not mentioned how the materials without hydrothermal pretreatment were prepared. 

The authors do not explain why they only show the Zeta potential measurements for the two samples the activated carbon treated with 0% KMnO4 HTP and prepared at 350 ï‚°C and the activated carbon treated with 1% KMnO4 HTP and prepared at 350 ï‚°C.

The notation of the four sets of experiments is a little confusing. Initially, they are numbered as

  1. Aquaculture system without a hydroponics system for control experiment (Figure 1a)
  2. Hydroponic system without connection to aquaculture system (Figure 1b)
  3. Aquaponics system without activated carbon filter (Figure 1c)
  4. Aquaponics system with 3 kg activated carbon filter (Figure 1d)

Then they are numbered as  pond 1, hydroponic system, pond 2 and pond 3 when the pH and the dissolved oxygen in water and the size of fish is discussed, and as Crop tube panel numbers 1,2,3, that correspond to hydroponic system, aquaponics system and aquaponics system with activated carbon filter respectively. This makes the manuscript hard to fallow.

Move Table 1 in section 3.2; Figure 3 in section 3.3 ; Figure 4 in section 3.5; Table 2 in section 3.7; Figure 7 in subsection 3.8.1; Figure 8 in subsection 3.8.2.

In Figure 6, crop the part of the SEM images with the technique’s details and add a 10µm scale bar in the photos for a better understanding of the strutures. 

Keep the same color coding in all the figures.

Author Response

I have edited as you suggested as attached.

Reviewer 2 Report

The reviewed manuscript describes the preparation of carbon materials from  Surgane leaf with hydrothermal process and their potential use as adsorbent for water treatment. In général, the experiments, results and their discussion included in the manuscript are presented in convincing manner. I do, however, have some questions, remarks and suggestions which should be considered in order to improve the manuscript prior its publication. They are listed below: 

  1. In the abstract part, authors need to present just the work realized and the relevant results obtained.
  2. For the introduction part, a comprehensive review is needed. For example, its better to present just the important information needed for the discussion of the results and also the bibliography of realised investigations in this area.
  3. Last part of introduction, from line 88 to 97. I think this part is better to be in materials and method part.
  4. .Also its necessary to introduce some bibliographic review on other raw materials used for the preparation of carbon materials and the relevant results obtained for the same application.
  5. In the Materials and methods part, all details should be presented, for example, the authors need to explain the method used for the preparation also more details about the treatment process.
  6. Can the authors explain the choose of temperature activation of samples. I think for the preparation of activated carbon with very developed surface area and porosity which is essential for the treatment of water, its necessary to use a higher activating temperature.
  7. In the results and discussion part, I think it’s necessary to present this part with convincing manner, starting by textural characterisation, chemical characterisation and morphological characterization.
  8. Also, it’s interesting to present each part with the corresponding figure, all figures must be presented in the corresponding part.
  9. 3.2 elemental composition, authors say that O/C ratios of pyrolyzed samples decrease with increasing temperature. If we examine the results present in the table1 we can see that is not the trend. For example, from 300C to 350C, O/c ratios increase for all samples.  Also, the % oxygen content doesn’t have the same trend for sample with 5% of KMnO4 and 0%. I suggest to reformulate this paragraph with homogenous conclusion.
  10. legends of the table 1 must be reformulated. w/t HPT is without hydrothermal pre-treatment . and not treatment.
  11. From the textural characterization its concluded that the sample prepared at 350C with 3% of KMnO4 is the best one. So, why not present all analysis of FTIR, XRD and zeta potential of this sample.
  12. Figure 2. Figure must be presented better; axis and graphs are not homogenous.
  13. figure 5, must be presented in correct form, especially legends.
  14.  3.5-part, paragraph explained the figure 5 and not 3.
  15. In the part of SEM characterization, I hope that authors can present images with the same magnification in order to see the effect of preparation conditions on the morphology.
  16. In the table of BET, authors need to present the textural characterization of samples at higher activation temperature. Because with temperatures used, we can’t see the difference between samples, also not enough to obtain sample with higher surface area.
  17. from all results presented, we can see that the sample prepared at 400C with 3% KMnO4 has the higher surface area with higher carbon content, why authors choose other sample to realise the water treatment experiments?
  18. from the above-mentioned comments, I suggest to reformulate this work and to present others new experiments which can give more interesting results. also, revision of some orthographic errors must be realised.

Author Response

I have made some modifications as you suggested, as attached.

Round 2

Reviewer 2 Report

Thank you for responding to all remarks presented in the first report . so, I agree with the publication of this paper in this form.

Congratulations

Sincerely Yours 

A Elmouwahidi